# Dynamic perceptive compensation for the rotating snakes illusion with eye tracking

**Yuki Kubota**[1]*, **Tomohiko Hayakawa**[2], **Masatoshi Ishikawa**[2]

**1** Graduate School of Information Science and Technology, The University of Tokyo, Bunkyo-ku, Tokyo, Japan, **2** Information Technology Center, The University of Tokyo, Bunkyo-ku, Tokyo, Japan

* yuki_kubota@ipc.i.u-tokyo.ac.jp

## Abstract

This study developed a dynamic perceptive compensation system for the rotating snakes illusion (RSI) with eye tracking. Large eye movements, such as saccades and blinks, were detected with an eye tracker, and perceptive compensation was dynamically performed based on the characteristics of RSI perception. The proposed compensation system considered three properties: spatial dependence, temporal dependence, and individual dependence. Several psychophysical experiments were performed to confirm the effectiveness of the proposed system. After the preliminary verification and determination of the temporal-dependent function for RSI perception, the effects of gaze information on RSI control were investigated. Five algorithms were compared using paired comparison. This confirmed that the compensation system that took gaze information into account reduced the RSI effect better than compensation without gaze information at a significance threshold of $p < 0.01$, calculated with Bonferroni correction. Some algorithms that are dependent on gaze information reduced the RSI effects more stably than still RSI images, whereas spatially and temporally dependent compensation had a lower score than other compensation algorithms based on gaze information. The developed system and algorithm successfully controlled RSI perception in relation to gaze information. This study systematically handled gaze measurement, image manipulation, and compensation of illusory image, and can be utilized as a standard framework for the study of optical illusions in engineering fields.

**Data Availability Statement:** All relevant data are within the manuscript and its Supporting information files.

## Introduction

It is widely known that physical light that reaches the receptive fields of the retina is processed by the brain to obtain perceptual information. This process often generates differences between the physical and perceived images. Discrepancies are manifested as optical illusions, including illusions of geometry [1], of color constancy [2], and of motion [3]. In recent years, various psychological research has been undertaken to elucidate the functions of various perceptions [4–6]. Additionally, research in information engineering to develop image presentation techniques that actively use human perceptual characteristics has been done [7, 8]. Studies on optical illusions reveal the mechanism of human perceptual function and support the

**Funding:** The authors received no specific funding for this work.

**Competing interests:** The authors have declared that no competing interests exist.

development of an augmented perception system that extends visual abilities and image presentation methodology based on human perception [9, 10].

Some optical illusions, including the rotating snakes illusion (RSI) [11], depend on the gaze information of observers, which includes eye position and eye movements (i.e., saccades and blinks). In the perception of the RSI, the intensity of the illusion increases in the peripheral vision and during large eye movements [12–14]. Thus, the RSI depends on spatial and temporal qualities of the gaze.

Previous studies investigated the perception of illusions dependent on gaze information. However, many of these studies conducted experiments using fixation points [14, 15]. Otero-Millan et al. indicated a relationship between microsaccades and blinks and the perception of the RSI using an eye tracking device [16], although eye trackers have not been used in feedback control for these effects. Recently, Chouinard et al. reported that the perception of vertical and horizontal illusion, which is a geometrical illusion, was different in relation to the presence and absence of gaze fixation [17]. The study concluded that the strength of the illusion increases in more stable retinal images with eye fixation, owing to a framing effect. Therefore, it is necessary to consider the spatial and temporal dependence of gaze information for controlling these illusions, including RSI.

Our study investigated means of reducing the effects of RSI perception in relation to eye movements. In particular, we addressed the question of whether it is possible to reduce the perception of the RSI in gaze-free conditions through dynamic compensation based on the management of gaze information. First, we designed and evaluated a dynamic perceptive compensation system and an algorithm to be synchronized with eye movements. After preliminary verification (Experiment 1) and determination of a temporally dependent function for RSI perception (Experiment 2), we investigated the effects of gaze information on RSI control (Experiment 3). In our experiments, we compared RSI images with compensation with gaze information, compensation without gaze information, and without compensation. The results indicated that RSI perception was reduced by our system that took the spatial, temporal, and individual dependence of the gaze information into consideration. Whereas, the compensation algorithm that took into account the spatial and temporal dependence of RSI perceptions had a lower score than other algorithms using gaze information. The results statistically confirmed that our compensation system, which took gaze information into account, reduced the RSI effect more appropriately than a compensation without gaze information, with $p < 0.01$ calculated with Bonferroni correction.

The technology of controlling gaze-dependent illusions is not limited to understanding the dynamics of visual perception. It can also be applied to eliminating the discrepancies in visual perception that are caused by optical illusions in the fields of virtual reality (VR) and human-computer interaction (HCI). By intervening in human perception by compensating for gaze-dependent illusions using a head-mounted display or glasses with an eye tracker, it may be possible to produce a system that make one aware of necessary perception and reduces unnecessary illusions. Because the preception of RSI alters with eye movements, use of this technology means we can achieve the fundamental knowledge necessary to intervene in human perception in relation to gaze information. Our research systematically handled gaze measurement, image manipulation, and compensation for an illusory image, and may be usable as a standard framework for research on optical illusion in engineering fields.

## Related work

The RSI shown in Fig 1 presents an illusory rotation in a still image [11], based on the Fraser–Wilcox illusion [3, 18]. This illusion comprises four gray-scale patterns: black, light gray,

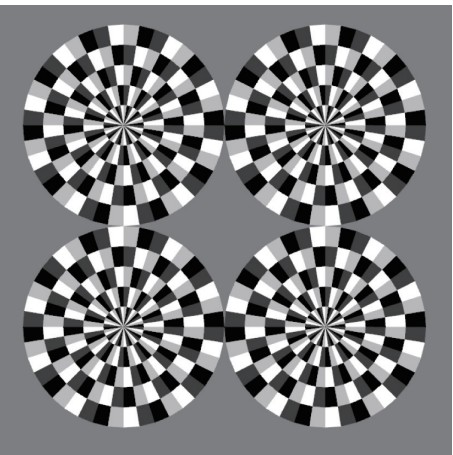

**Fig 1. Stimulus patterns that perceive the RSI.** This illusion incorporates four gray-scale patterns: black, light gray, white, and dark gray. The direction of motion changes depending on the arrangement of the pattern. Our study repeatedly aligned the patterns to enhance the intensity of the illusion.

white, and dark gray. The direction of the motion changes depending on the arrangement of the pattern. Our study repeatedly aligned the patterns to enhance the intensity of the illusion.

The intensity of the RSI increases in peripheral vision [13], and the following two mechanisms for the RSI in relation to eye movement have been produced in previous studies [12, 14, 19].

1. **Difference in adaptation time after refreshing retinal images**: the illusory motion is perceived by the adaptation time difference in brightness after suppression occurs through large eye movements.

2. **Update of retinal images due to small eye movements**: the illusory motion is perceived due to updates to retinal images with constant small eye movements such as tremors and microsaccades.

The former hypothesis holds that illusory motion occurs during a convergence of visual processing [12, 13]. In other words, the differences in the timing of the disappearing afterimages in a certain location on the retina cause apparent motion. The latter hypothesis holds that the motion of retinal images entering the visual system generates an asymmetrical motion signal [14]. Backus and Oruç measured the temporal dependence of RSI perception by updating an image, and the results indicated the existence of two mechanisms, caused by large and small eye movements. Our study mainly considers the effects of large eye movements for RSI control.

Other studies have reported that infants [20], fish [21], and cats [22] also perceive the RSI, and the intensity of the illusion decreases by age [23]. The effects of contrast [15], color [24], and afterimages [25] have also investigated in previous studies. In addition, a video prediction machine using deep neural networks has been found to perceive the RSI [26]. In this study, a machine designed by Watanabe et al. leaned a first-person viewpoint video taken at an amusement park and was tested to predict the direction of unlearned RSI images, and they succeeded in reproducing perception similar to that of humans, in which the motion is perceived or not depending on the given arrangement of RSI patterns.

## Proposed system and algorithm

### Dynamic perceptive compensation system synchronized with eye movement

This research proposed a system that dynamically compensates for optical illusions using gaze information. The schematic diagram of the system is shown in Fig 2(a). This system consisted of an eye tracker for measuring eye movement, a PC with an image-processing device, a display for presenting images, and a keyboard for subjects to enter their response. The gaze position of the subjects was acquired by the eye tracker and dynamically included a openFrameworks program running on the PC. The amount of eye movement was calculated from the gaze position, and the presented image was compensated based on the calculation by rotating or translating parts of the image. Driving the system dynamically made it possible to compensate for the visual illusion using gaze information.

### Compensation algorithm

A compensation algorithm was devised using the following three characteristics of RSI, as reported in previous studies [12, 13].

1. Large eye movements such as saccades and blinks temporarily increase the intensity of the RSI.

2. The RSI effect is larger in the peripheral vision than the fovea.

3. Large eye movements have greater effects than small eye ones in an environment where the eyes are actively moving.

The scheme of the compensation algorithm is given in Fig 2(b). Here, an eye tracker measured the eye position of the observers $p_{now}(X, Y)$, and the distance of eye movement $\Delta d$ was measured with the difference between the present and past frame. When $\Delta d$ exceeds its threshold $\Delta d_{th}$, the system judged that a saccade had occurred. When the eye position $p_{now}(X, Y)$ could not be detected, the system judged that a blink had occurred. The eye movement was determined every two frames despite the attendant increase error probability to improve the

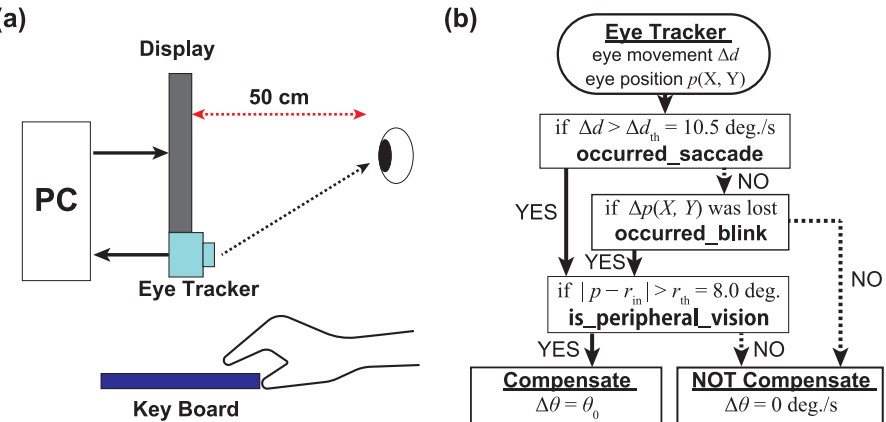

**Fig 2. Proposed system and algorithm.** (a) Schematic diagram of dynamic perceptive compensation system including an eye tracker for measuring eye movement, a PC with an image-processing device, a display for presenting the images, and a keyboard for subjects to enter their response. (b) Compensation algorithm applied to the RSI as the eye tracker detected saccades or blinks. Only the disk of RSI whose center existed in the peripheral visual field was rotated, with an angular velocity $\Delta\theta$ during the time $\Delta t$, to reduce the illusory effect.

real-time performance of feedback control. For example, when the system was driven at only 6 Hz, based on the average value of 15 frames at 90 Hz, it was difficult to dynamically measure the temporal dependence of eye movements.

When large eye movements occurred, the system determined where to compensate. If the distance between a central position for each disk $r_{in}$ and the present eye position $p_{now}(X, Y)$ was larger than a threshold $r_{th}$, the disk was considered to locate in the peripheral vision. Only the disks located in the peripheral vision were physically rotated by the system at a constant angular velocity $\Delta\theta = \theta_0$ during compensation time $\Delta t$.

This algorithm had four ad hoc parameters: a saccade threshold $\Delta d_{th}$, a peripheral vision threshold $r_{th}$, a compensation time $\Delta t$, and a compensation angular velocity $\Delta\theta$. In Experiment 1, $r_{th}$ and $\Delta t$ were determined with reference to previous studies [12, 13], and were set to $r_{th}$ = 8.0 deg deg. and $\Delta t$ = 300 ms. $\Delta d_{th}$ was determined as 10.5 deg./s in our pilot study, which determined the threshold for distinguishing whether an occurrence was saccades or not. Furthermore, $\Delta\theta$ was changed as the experimental parameter. In Experiment 2, in relation to the temporal dependence of RSI perception, $\Delta t$ was set as the measurement variable for the experiment. In Experiment 3, considering both the spatial and temporal dependence of the perception, $\Delta t$ and $r_{th}$ were variables. The detailed settings for each experiment are discussed below.

## Experimental setting and stimuli

**Experimental setting.** Experiment 1 used a system that consisted of an eye tracker at 90 Hz (Tobii Eye Tracker 4C), a laptop PC with a keyboard (ThinkPad T440p, Intel(R) Core i-7-4710 MQ 2.50GHz), and a display at 60 Hz (BENQ G2411HD, 1920 × 1080). Experiments 2 and 3 used the system incorporating an eye tracker (Tobii Eye Tracker 4C) driven at 90 Hz, a desktop PC (Dell, Precision 7920 Tower, Intel(R) Xeon(R) Silver 4210 2.20GHz), and a display at 60 Hz (BENQ G2411HD, 1920 × 1080). In all experiments, the eye tracker itself was driven at 90 Hz to detect the eye position, and the computation time for each processing loop should be kept within approximately 10 ms to drive the whole system at 90 Hz. However, a bottleneck in system processing caused a delay of more than 10 ms. According to data from our pilot study (S1 Dataset), the computational loop from the detection of the eye movements to the image display took 20.3 ± 1.1 ms (approximately 50 Hz), and the system took this delay as a bottleneck when the devices were synchronized.

The sampling rate for our eye tracker was below the 1000 Hz benchmark achieved by research-grade eye trackers, and previous studies have found that degradation in the number of detected saccades and inaccurate estimations of saccade duration occurred with eye trackers that have low sampling rates, although accurate measurement of fixation times and points is possible with such trackers [27, 28]. With our eye tracker, it was difficult to follow the saccade process in detail, and the tracker sometimes failed to detect brief saccades, which led to a failure of compensation based on the detection of these brief saccades. On the other hand, our algorithm, as described above, did not require direct measurement of the number or duration of saccades, so it was assumed that the proposed algorithm was able to work adequately through detecting large eye movements and blinks.

For all of the experiments, the distance between the subject and the display was fixed at 50 cm using a chin rest. The center of the head was aligned to the center of the display. Earmuffs were worn to eliminate audio interference. All subjects had normal or corrected-to-normal vision. All experiments were conducted in a constant room lighting environment, and the background brightness of the screens as measured with a brightness meter (Konica Minolta, Color Luminance Meter CS-150) was 30 cd/m$^2$. Although there was an asymmetry in the

**(a)**                                              **(b)**

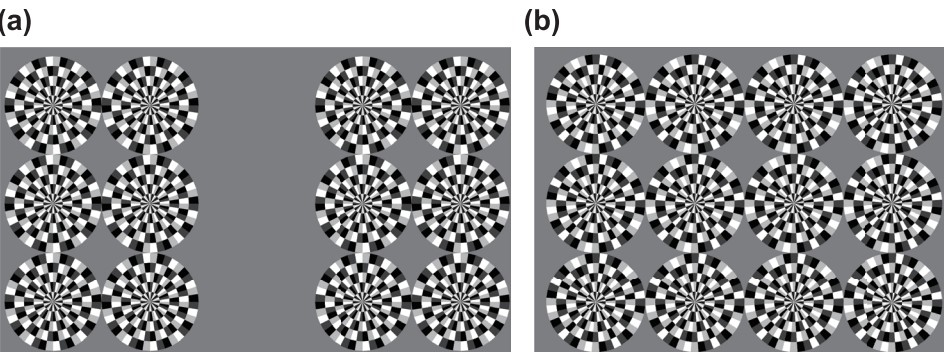

**Fig 3. Experimental stimuli.** (a) Images used in Experiments 1 and 3 (Clockwise image). (b) Images used in Experiment 2 (Control condition). All stimulus images consist of RSI disks with four 8-bit grayscale colors: black, dark gray, white, and light gray. The arc of each disk is 9.0 deg., and its diameter is 150 px. Two sets of six disks (Experiments 1 and 3) and one set of 12 disks (Experiment 2) are used as experimental stimuli.

reflection or glare of the room light on the display, the effect was eliminated because which of the two images were given on a given side of the display was randomized in all experiments.

**Experimental stimuli.**   The images used in the experiment are shown in Fig 3. Fig 3(a) shows an image used in Experiments 1 and 3 and Fig 3(b) in Experiment 2.

All stimulus images consisted of some RSI disks that comprise four 8-bit gray-scale colors: black (pixel number: 0, brightness: 4 cd/m$^2$), dark gray (pixel number: 63), white (pixel number: 255), and light gray (pixel number: 191). The arc of each disk was 9.0 deg., and the diameter of each disk was 150 px (4.73 deg.). Two sets of six (two × three) disks (Experiments 1 and 3), and one set of 12 (four × three) disks (Exp. 2) were used as experimental stimuli. The former image is shown in Fig 3(a), and the latter is shown in Fig 3(b). In the former, the inner disks were horizontally placed 300 px from the center of the screen, and the outer disks were placed 300 px further from the inner disks. In the vertical direction, the central disks were aligned with the centeral axis of the display, and the upper and lower disks were placed 300 px above and below the center disks. In the latter, the center disks were placed horizontally at a distance of 150 px from the center of the display and vertically at the center of the display. The image disks had three horizontal rows and four vertical columns and circumscribed each other. That is, the distance between the centers of the disks was 300 px in both the horizontal and vertical directions.

Experiment 1 used stimuli of clockwise rotation (CW image) and counterclockwise rotation (CCW image). The CW image was also used in Experiment 3. Experiment 2 used an illusory image, for which the observers perceived CW rotation (illusion condition) and the image for which observers did not perceive any movement (control condition).

## Experiment 1: Preliminary verification and parameter estimation of the proposed system

This section presents the experiment for preliminary verification and the optimal parameter estimation for the compensation angular velocity. The experimental procedures with paired comparison and the results of the experiment are described as follows.

### Experimental procedures and conditions

Nine subjects, aged 22—32 years, observed the stimuli shown in Fig 3(a) for approximately ten seconds and chose the stimulus that was perceived to move to a greater extent. The

participants were required to choose a stimulus even if they could not distinguish between them with certainty. One set of the six disks was chosen at random to be rotated by the system and the algorithm, and the other remained just still images. The subjects recorded their responses using the left and right arrow keys on the keyboard. No eye fixation point was prepared, because the experiment measured the effects of the eye movements. It should be noted that the subjects gave their responses only once per task after observation of the stimuli; the subjects probably responded their perception, which of the two image sets had moved to a great extent, after considering all of their observation during a task (all data are available as S3 Dataset and examples of the gaze heat map obtained in our experiment are shown in S1 Fig to S3 Fig).

The experiment had two levels and 18 conditions: stimulus images (the patterns causing CW and CCW motion; two conditions) and compensation angular velocity (−2.0 to 2.0 deg./s by 0.5 deg./s; nine conditions). Each condition has ten tasks for every subject after ten training tasks. All tasks were presented at random for each subject.

The analysis was performed using the Friedman test, the Tukey–Kramer method, and the Steel–Dwass method. The selection rate (SR) used to analyze the results was calculated as follows. For each subject, the SR for ten tasks under each compensation angular velocity condition was calculated: the SR of 0.6 was considered if the subject selected the compensated image six times in ten tasks. The SR for each subject was used as the statistical parameter. The threshold of significance was set at $p < 0.05$.

The experiment followed the tenets of the Declaration of Helsinki and were approved by the Ethics Committee of the University of Tokyo (Approval Number: UT-IST-RE-181107-1).

## Results

The experimental results are shown in Fig 4. The horizontal axis indicates the compensation angular velocity of $\Delta\theta$, and the vertical axis indicates the average of the SR for subjects to choose the compensated image. Lower SRs indicate more effective reduction of the RSI effect by the system. The blue line denotes the results for the CW images, and the red line gives the results for the CCW images. The original data are available in the Experiment 1 Table in the (S2 Dataset).

The results indicate that an average of the SR for the CCW images rotated at 0.5 deg./s was 0.322, and the average for the CCW images rotated at −0.5 deg./s was 0.411, both of which are smaller than 0.544, the rates for both of the correspondent control conditions. On the other hand, the average of SR for the CCW images rotated at −0.5 deg./s was 0.956 and that for the CW images rotated at 0.5 deg./s was 0.900. That is, system compensation against the perceived rotation can reduce the RSI effects when the appropriate angular velocity is chosen: forming an approximate ± 0.5 deg./s rotation against perceived motion in our experimental design. Note that this optimal parameter is expected to vary in relation to experimental conditions such as the lighting environment, display brightness, and specific pattern arrangement.

Non-parametric analysis is conducted using the Friedman test because the results of the Shapiro–Wilk test for each condition indicate that some data cannot be assumed to be the normal data. A Friedman test for both the CW and CCW conditions revealed significant effects of angular velocity for the CW ($\chi^2 = 58.7$, $p < 0.001$) and the CCW ($\chi^2 = 61.63$, $p < 0.001$) condition. These results suggested the existence of an optimal parameter for RSI reduction. Below, the data for the control condition ($\Delta\theta = 0$ deg./s) and for the conditions of $\Delta\theta = \pm 0.5$ deg./s were compared. A Tukey–Kramer test indicated that the results for the CCW images that rotated at 0.5 deg./s were significantly smaller than those for the controlled condition

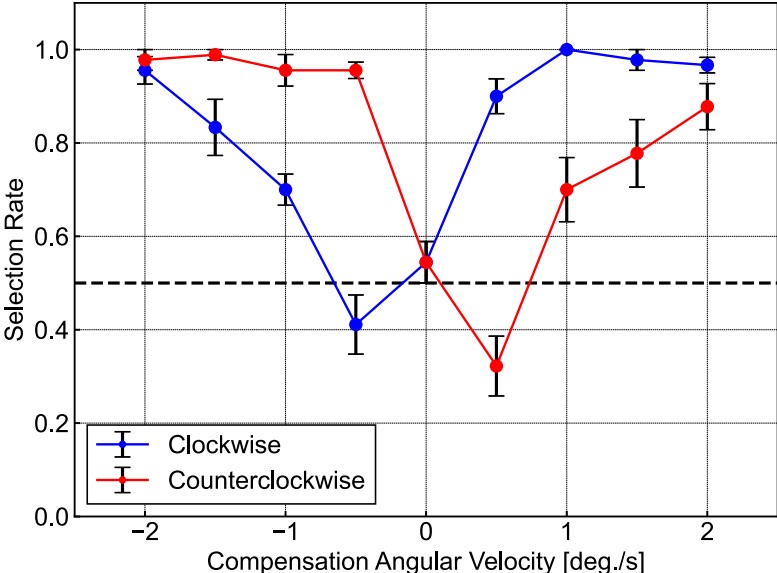

**Fig 4. Results of Experiment 1.** Each value indicates results averaged across the subjects. The horizontal axis presents the compensation angular velocity for $\Delta\theta$, and the vertical axis indicates the average of the selection rate (SR) for subjects for choosing the compensated image. Here, the lower the SR, the more effective reduction of the RSI effects by the system. The blue line denotes the results for the CW images, and the red line gives the results for the CCW images. The original data are available in the Experiment 1 Table of the (S2 Dataset).

($p = 0.0063$) and the CW images that rotated at −0.5 deg./s had a smaller trend ($p = 0.16$). Moreover, a significantly asymmetric SR was observed for the same absolute angular velocity 0.5 deg./ s for the CW ($p < 0.001$) and for the CCW ($p < 0.001$) images, which indicated that the system's compensation asymmetrically affected the RSI perception. Note that the data for the CCW images at $\Delta\theta = 0.5$ deg./s cannot be assumed to be normal data points, and some of the data had a limited range of assumption for normality. A Steel–Dwass test found the data for the CCW images at 0.5 deg./s had nearly significantly smaller value than that for the controlled condition ($p = 0.058$), and the CW images at −0.5 deg./s did not have significant effects ($p = 0.278$).

Consequently, this preliminary experiment suggested that the intensity of the RSI could be reduced by the proposed system. Specifically, the compensation angular velocity of 0.5 deg./s against the perceived motion can reduce the effects of the RSI for both the CW and CCW images in these experimental settings, although the parameter may vary in relation to the environmental conditions, such as display brightness and lighting. As previously described, this study investigated the possibility of perceptual compensation in situations similar to those of everyday environments by acquiring data under gaze-free conditions. Considering the display size used in this study, the system was able to compensate even when the eyes moved freely on average within a viewing angle range of 45.1 deg. (1500 px) horizontally and 28.0 deg. (900 px) vertically taking into consideration the display range of RSI disks, though it is possible that this compensation could work more or less easily depending on the viewpoint of the subjects. For example, in an environment with significantly different display luminance due to non-uniform illumination, the optimal parameters may differ depending on viewpoint, which would cause a dependence of the viewpoint on system effectiveness.

## Experiment 2: Determination of the temporal-dependent characteristics of the RSI

Experiment 1 indicated the existence of optimal parameters for the reduction of RSI rotation, and it was suggested that the proposed system could reduce RSI effects. This section investigated temporally dependent functions, whose importance will be discussed in Experiment 3. The method of sinusoidal stimulation was introduced, and then the procedures and results of the experiment for determining temporally dependent functions were established.

### Use of sinusoidal stimulation

The compensation algorithm given in Experiment 1 had a constant parameter for $\Delta t$. This experiment adopted sinusoidal stimulation as $I = I_0 \cdot \sin(\pi t/\Delta t)(0 \leq t \leq \Delta t)$, otherwise $I = 0$, where $I_0$ is the maximum value of the stimulus, $t$ is the elapsed time, and $\Delta t$ is a specific scale for the compensation time. The system rotated each disk at a half cycle of a sinusoidal variance after the saccades or blinks were detected. This method featured a single peak for evaluating the temporal dependence of the illusory perception, which was more appropriate, owing to the continuous changes in the stimulus parameter.

This algorithm had three ad hoc parameters: the threshold for saccade distance $\Delta d_{th}$, the compensation time $\Delta t$, and the compensation angular velocity $\Delta \theta$. The threshold was set at $\Delta d_{th}$ 10.5 deg./s, the same value as Experiment 1. $\Delta t$ and $\Delta \theta$ were the measurement variables in Experiment 2.

### Experimental procedures and conditions

Seven subjects aged 21—33 years observed the stimulus image for approximately ten seconds per each task and reported whether the image had rotated right or left. They were instructed to choose one of the two responses even if they could not select one with certainty. The subjects gave their responses using the keyboard. The angular velocity $\Delta \theta$ was updated using parameter estimation by sequential testing (PEST) taking into account the subjects' response, in an adaptive-method PEST [29] that updated the parameter based on heuristic rules. In this experiment, the PEST had an initial value of $\pm$ 10.0 deg./s, an initial update width of 2.0 deg./s, and end condition of 0.0625 deg./s. Using the PEST, the point of subjective equality (PSE) was obtained for each condition. Furthermore, the experiment was performed using the interleaved staircase method for the subjects to avoid predicting the stimulus transition. The subjects began the actual tasks after performing approximately 20 training tasks.

The experiment had 12 conditions: stimulus image (the illusory image and the non-illusory image; two conditions) and compensation time $\Delta t$ (100 ms, 150 ms, 250 ms, 500 ms, 1000 ms, and $\infty$, where $\infty$ indicates rotation at a constant angular velocity, regardless of eye movement; six conditions). All tasks were presented at random for each subject. Counterbalancing was performed by blocks of tasks grouped by stimulus image. In other words, half of the subjects were randomly presented with a task for each $\Delta t$ condition for the illusory image, and after all of the tasks with the illusory image were completed, they were randomly presented with a task in the control condition. The other half received the tasks in the reverse order.

The analyses were performed using the Friedman test, the Tukey–Kramer method, and the Steel–Dwass method. The calculation of the statistical values and the analyses of the results were performed on 14 data points for seven subjects, obtained with the interleaved staircase method. The significant threshold was set at $p < 0.05$.

The experiment followed the tenets of the Declaration of Helsinki and were approved by the Ethics Committee of the University of Tokyo (Approval Number: UT-IST-RE-181107-1).

## Results

The results are shown in Fig 5, where the horizontal axis indicates compensation time $\Delta t$, the vertical axis gives the average PSE of the angular velocity, as obtained by PEST, across all of the subjects. The red and blue lines denote the results for the illusory and control conditions, respectively. The vertical dashed line indicates $\Delta t = 500$ ms and the error bar indicates the standard error. Note that the data for $\Delta\theta = -8.69$ deg./s, which were obtained as the estimated PSE angular velocities of one subject for $\Delta t = 500$ ms, were eliminated from our analysis and replaced with an average of the other subjects' results, due to an outlier assessed using the Smirnov-Grubbs test ($p = 9.77 \times 10^{-4} < 0.001$). The data for $\Delta t = \infty$ are alternatively plotted at $\Delta t = 5000$ ms in Fig 5. The original data are available in Experiment 2 Table in the (S2 Dataset).

The graph indicates that the results for the illusory condition had a negative PSE for all $\Delta t$, which reproduces the results of Experiment 1, indicating that compensation in the opposite direction can reduce the intensity of the RSI. The absolute value for PSE was largest at 100 ms ($\Delta\theta = -2.56$ deg./s) and the value for $\Delta t = 500$ ms was less than half ($\Delta\theta = -0.830$ deg./s) of the value of $\Delta t = 100$ ms. The temporally dependent function for RSI perception was appropriately obtained, used for Experiment 3, which reflected the temporal effects of gaze information on RSI motion reduction:

$$
\begin{aligned}
f(t\,[\mathrm{ms}]) \quad &= 2.56\sin\left(\pi t/100\right) + 1.66\sin\left(\pi t/150\right) + 1.80\sin\left(\pi t/250\right) \\
&+ 0.830\sin\left(\pi t/500\right) + 0.554\sin\left(\pi t/1000\right) + 0.277
\end{aligned}
\tag{1}
$$

Non-parametric analyses using the Friedman test were performed following the results of the Shapiro–Wilk test. The Friedman test indicates that the illusory condition featured a significant effect of compensation time ($\chi^2 = 34.5$, $p < 0.001$), while the control condition had a

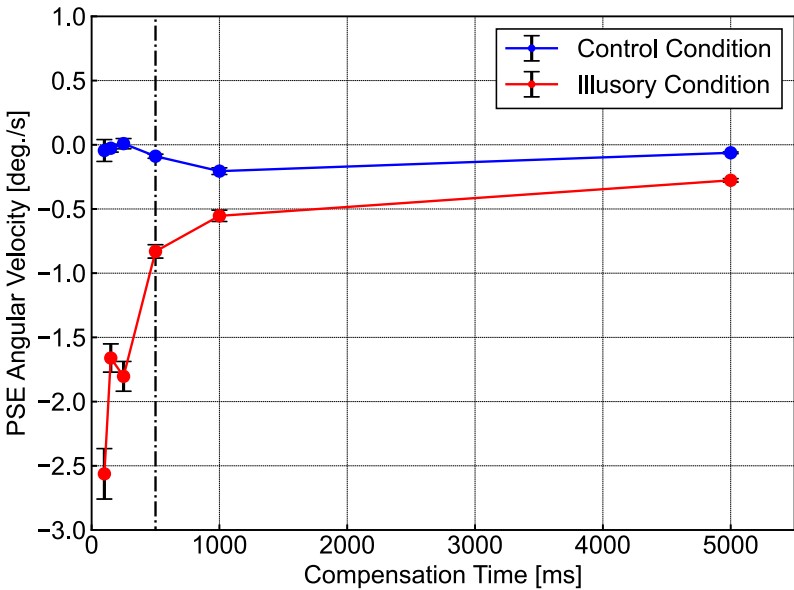

**Fig 5. Results of temporal dependence of RSI perception.** The horizontal axis presents the compensation time $\Delta t$, and the vertical axis gives the average PSE for the angular velocity obtained by PEST across the subjects. The red and blue lines denote the results for the illusory and control conditions, respectively. The vertical dashed line indicates $\Delta t = 500$ ms, and the error bar presents the standard error. The data for $\Delta t = \infty$ are alternatively plotted at $\Delta t = 5000$ ms. The vertical dashed line indicates $\Delta t = 500$ ms. The original data are available in Experiment 2 Table in the (S2 Dataset).

limited effect ($\chi^2 = 2.45$, $p = 0.784$). This suggests that the perception of the RSI image had temporal dependence. Below, the results of the Tukey–Kramer test are presented, because all results of the Shapiro–Wilk test for the illusory condition suggest an absense of the normality deviation. The results between the PSE for $\Delta t = 100$ ms and those for other conditions show that the absolute value of PSE for $\Delta t = 100$ ms was significantly larger than those for $\Delta t = 500$ ms ($p = 0.0185$), and $\Delta t = 1000$ ms ($p = 0.0036$), $\Delta t = \infty$ ($p < 0.001$). These results qualitatively reproduced the results of previous studies [12], and the RSI motion compensation was effective up to 500 ms after the occurrence of large eye movements.

On the other hand, almost all of the PSEs of the control condition had negative values, except for the results for $\Delta t = 250$ ms, which could be explained by potential selection bias in the direction of rotation: half of the subjects first worked on the illusory condition task block, and observed an illusory image due to the counterbalancing performed with the blocks of tasks groped by stimulus images. In other words, some subjects continued to observe an illusory image during the first task block, which may have biased their perception of perceived rotation when observing the control image.

## Experiment 3: Evaluation experiment of gaze information effects on RSI reduction

In the previous experiments, we determined the appropriate parameters and temporally dependent functions for RSI perception. However, parameter optimization remained as a necessity for the precise reduction of the RSI motion, as RSI differs across individuals. Furthermore, it was considered important to establish the effects of gaze information on RSI reduction to determine the most effective algorithm to acheive reduction. Therefore, this section describes the evaluation experiment that verifies the effects of gaze information on RSI motion reduction. We introduced five algorithms, with and without gaze information, and evaluated the effectiveness of reducing the effects of RSI in the experiment.

### Overview of compensation algorithms

This experiment compared five compensation algorithms, as follows:

**Algorithm O, no compensation**: An image was presented as a still image, without compensation.

**Algorithm A, spatially dependent compensation**: An image was presented with spatially dependent compensation that changed its angular velocity depending on the viewing angle.

**Algorithm B, temporally dependent compensation**: An image was presented with temporally dependent compensation that changed the angular velocity depending on the time elapsed after large eye movements were detected.

**Algorithm C, spatially and temporally dependent compensation**: An image was presented with spatially and temporally dependent compensation that changed angular velocity depending on the viewing angle and the time elapsed after large eye movements were detected.

**Algorithm U, uniform compensation**: An image was presented with uniform compensation that had a constant angular velocity, independent of gaze information.

The spatially dependent compensation linearly increased angular velocity toward the peripheral vision, as in Fig 6(a). While the algorithm described in Fig 2(b) had a threshold for the peripheral vision and was switched on and off there, the method featured here changed the

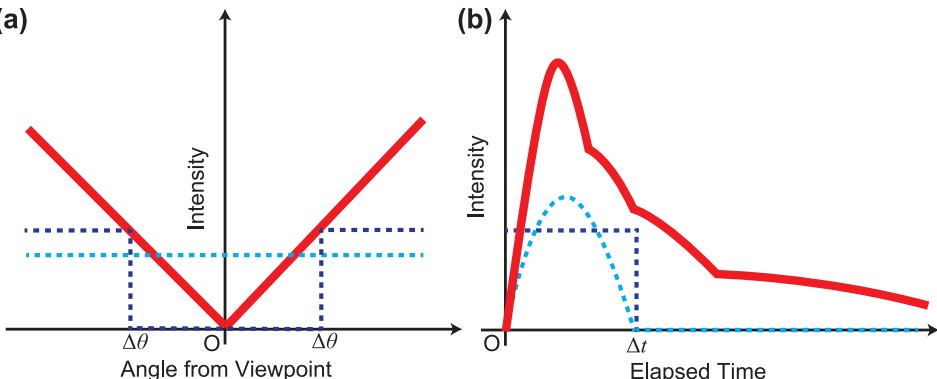

**Fig 6. Schematic diagram of compensation function.** (a) Spatially dependent function. (b) Temporally dependent function. The red line denotes the function used in this experiment, and the dark and light blue dashed lines denote the function used in Experiments 1 and 2, respectively.

angular velocity in a linear fashion depending on the viewing angle to eliminate the discontinuity of the compensation region. That is, the compensation angular velocity $v(\theta)$ was $c_A \cdot \theta$, where the viewing angle was $\theta$. In addition, the angular velocity was uniform for each disk of the RSI, to consider the consistency of changes in the imagery. Here, $\theta$ was determined from the center position of each disk.

The temporally dependent compensation changed angular velocity only after large eye movements were detected, which was based on the temporal-dependent function in Experiment 2. The temporally dependent function of RSI $f(t)$ was obtained as Eq 1, as shown in Fig 6(b): $f(t) = \sum_i a_i \sin(\pi t/\Delta t_i)$, where $a_i$ is the $i$-th value of coefficient, $t$ is the elapsed time, and $\Delta t$ is the specific scale for the compensation time. Note that each sinusoid component must be zero if $t > \Delta t_i$. The experiment tuned the entire intensity of the temporally dependent function for each individual: $v(t) = c_B \cdot f(t) = c_B \cdot \sum_i a_i \sin(\pi t/\Delta t_i)$. Furthermore, the elapsed time was reset to $t = 0$ every time a large eye movement was detected to establish the recurrence of the perception of the RSI.

The spatially and temporally dependent compensation is designed to combine the two algorithms above: $v(t, \theta) = c_C \cdot \theta f(t) = c_C \cdot \theta \sum_i a_i \sin(\pi t/\Delta t_i)$. The uniform compensation rotated all RSI disks at a constant angular velocity, regardless of eye movement: $v = c_U$. The proportionality constants $c_A$, $c_B$, $c_C$, $c_U$ were determined for each subject using the adjustment method.

## Experimental procedures and conditions

In this experiment, the proportionality constants for each method were first tuned for each individual, and then five compensation algorithms were compared, using paired comparison. Seven subjects aged 21—33 years participated in this experiment.

**Experiment 3-1: Calibration step.** Previous studies have reported that the perception of the RSI varies with the individual due to the effects of aging [23], image contrast [15], and other factors. Thus, the compensation parameter should be personalized. In this step, the proportionality constants for each method were determined using the adjustment method for each subject. The subjects observed the stimuli shown in Fig 3(a). They were instructed to change the parameters using the keyboard and identify the parameters for which the image was perceived as the most stationary. The subjects were forced to select the optimal parameter even if they felt that the motion had not completely ceased. As the parameters changed, all of the disks in the images were rotated by the system and the algorithm. The fixation point and the answer time limits were not set due to the adoption of the adjustment methods. Each parameter was adjusted by

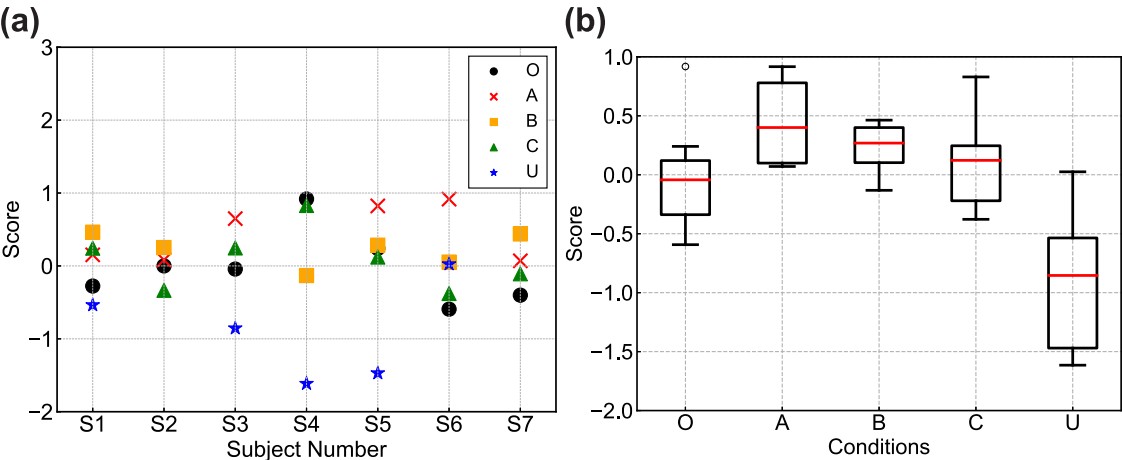

**Fig 7. Individual results for evaluation values of the paired comparison in Experiment 3-2.** (a) Individual results. (b) Box plot of the evaluation values between subjects. The data that are presented with a circle in the results for Algorithm O are shown as an outlier because they are more than 1.5 times the interquartile range from the third quartile. The original data are available in the Experiment 3-2 Table in the (S2 Dataset).

0.025. This stage was performed twice for each subject and condition. In total, eight tasks (two for each of four coefficients) were randomized for each subject. The average of these two adjusted parameters was adopted as the coefficient in Experiment 3-2.

**Experiment 3-2: Evaluation step in the paired comparison.** The effectiveness of the compensation was evaluated with the paired comparison, based on the obtained coefficient in Experiment 3-1. The subjects observed the stimuli as seen in Fig 3(a) and selected those that appeared to a greater extent in one task. Two algorithms were randomly selected from the five algorithms described above to rotate the left and right disk sets. The coefficients used in this experiment were those obtained in Experiment 3-1. The experiment was performed for 10 conditions 12 times each, in a combination of five methods. All tasks were presented at random.

Experimental data showing that the proportional constant of the algorithm was precisely zero were regarded to indicate a still image (Algorithm O) in our analysis. The score values used for evaluation were calculated as follows. The score for each subject shown in Fig 7 was calculated from subject's selection ratio for each paired comparison. The inverse of the standard normal distribution function was used to estimate the score corresponding to the selection ratio. Furthermore, the p-value shown in Table 2 were calculated using paired-comparison data for the sum of all subjects. The selection ratio and scores corresponding to it were calculated from the data. The p-value for each algorithm pair was estimated using the averages and standard deviations of the score values. Note that the individual data points for which the number of selections was (0, 12) were replaced with (0.5, 11.5) to avoid entering zero values into the normal distribution. This zero-value correction was not performed for all data that used for the box plot and p-value calculation for all data. Significance was set to $p < 0.05$.

The experiment followed the tenets of the Declaration of Helsinki and were approved by the Ethics Committee of the University of Tokyo (Approval Number: UT-IST-RE-181107-1).

## Evaluation results

From Experiment 3-1, the obtained parameter values for each algorithm were $c_A = -0.71 \pm 0.08$, $c_B = -0.35 \pm 0.05$, $c_C = -0.39 \pm 0.08$, and $c_U = -0.14 \pm 0.03$ where the error is the standard

**Table 1. The ratio of larger scores and quartiles of the results.**

| Algorithm | Ratio of larger scores | First quartile | Median | Third quartile |
|:---:|:---:|:---:|:---:|:---:|
| O | - | −0.307 | −0.043 | 0.120 |
| A | 6 / 6 | 0.117 | 0.400 | 0.780 |
| B | 5 / 6 | 0.153 | 0.269 | 0.400 |
| C | 4 / 7 | −0.164 | 0.122 | 0.246 |
| U | 1 / 5 | −1.316 | −0.854 | −0.536 |

The selection ratio shows the number of subjects' results with larger scores than those in the still image. The results for subjects who chose the parameter at 0 deg./s eliminated both the numerator and the denominator of the ratio, which suggests that the smaller denominator corresponds to a limited effect on the RSI reduction.

error (original data are available in the Experiment 3-1 Table of the S2 Dataset file). Looking at spatial dependence at $t = 100$ ms, the spatially dependent compensation (Algorithms A and C) had larger values than the spatially independent algorithm (Algorithms B and U), except near the fovea, in the range of ± 1.5 deg. On the other hand, Algorithm C produced a decreasing value until it reached $t = 1000$ ms, showing the same as in the uniform compensation. For the temporally dependent function, the rise and convergence of the RSI perception were reproduced by $f(t)$. Algorithms B and C had larger values up to 500 ms than the temporally independent compensation (Algorithms A and U). This reflects the results of Experiment 2, namely, the PSE at 500 ms was less than half of the PSE at 100 ms. Further, through executing a calibration step, the optimal algorithm parameters could be obtained for each individual, and each of the proposed algorithms was personalized.

Fig 7 shows the evaluation value for Experiment 3-2. The individual evaluation values were shown in Fig 7(a). The horizontal axis of Fig 7(a) denotes the subjects, and the vertical axis denotes the scores for the paired comparison. Here, larger scores indicated that compensation algorithms worked more effectively. Fig 7(b) shows the box plot of the evaluation values between subjects. The horizontal axis denotes the conditions, and the vertical axis the scores. The red line shows the median, and the ends of the box show the first and third quartiles. The data that are presented with a circle in the results for Algorithm O are outliers because they were more than 1.5 times the interquartile range from the third quartile. Table 1 presents a summary of the results, with the ratio of larger scores and quartiles for each algorithm. The selection ratio shows the number of subjects' results with larger scores than the still image. The results for subjects who chose the parameter at 0 deg./s eliminated both the numerator and denominator of the ratio, which suggests that a smaller denominator indicates a limited effect on RSI reduction. The original data are available in the Experiment 3-2 Table of the (S2 Dataset).

From these results, Algorithms A, B, and C had larger median values than Algorithm O, which presented just a still image. Algorithm U, which used the uniform compensation without gaze information, had smaller values than Algorithm O. These results suggested that gaze information is important for RSI reduction.

Table 2 shows the results for the p-values for each algorithm. Note that the Bonferroni correction should be considered when we examine the p-values. Algorithms A, B, and C had significantly larger score than Algorithm U at $p < 0.01$ with Bonferroni correction, which means that compensation with gaze information is important. Algorithm U had significantly lower values than Algorithm O, which presented just a still image. This indicates the compensation without gaze information could not function appropriately. Algorithms A and B had significantly higher scores than Algorithm O. This means that Algorithms A and B could reduce RSI perception more effectively than a still illusory image. However, Algorithms O and C did not

**Table 2. P-values calculated with the Bonferroni correction for the algorithms.**

| P-Value | Algo. O | Algo. A | Algo. B | Algo. C | Algo. U |
|---|---|---|---|---|---|
| O | - | $1.8 \times 10^{-11}$ | $2.2 \times 10^{-4}$ | 1.0 | $1.4 \times 10^{-25}$ |
| A | - | - | $8.4 \times 10^{-2}$ | $8.4 \times 10^{-7}$ | $2.8 \times 10^{-22}$ |
| B | - | - | - | $4.0 \times 10^{-2}$ | $5.2 \times 10^{-22}$ |
| C | - | - | - | - | $1.3 \times 10^{-24}$ |
| U | - | - | - | - | - |

The p-values are calculated with Bonferroni correction, and ten times the values before correction are presented in each cell.

show a significant difference. This may be due to a superposition effect; we give details on this in the discussion.

These results indicate that compensation incorporating gaze information is essential for reducing RSI effects under gaze-free condition.

## Discussion

This study proposed a method of performing dynamic perceptive compensation for RSI and verified its effectiveness using several experiments. The results are summarized below, along with guidance for developing a system of gaze-dependent dynamic illusion groups. Moreover, the possibilities and scalability of our results are also described.

### Effectiveness of dynamic perceptive compensation

In this study, dynamic perceptive compensation takes three properties into account: spatial dependence, temporal dependence, and individual differences. Spatial dependence describes the increase in RSI perception in peripheral vision, and temporal dependence indicates the increase in RSI perception immediately after large eye movements. Five compensation methods were compared according to their effectiveness in RSI motion reduction.

From the comparison experiments in Experiment 3, the following results and guidelines are summarized.

- **Effect of eye movement consideration**: It is essential to consider eye movement and peripheral vision for the reduction of optical illusion depending on eye movement. Both temporal and spatial dependence have a major effect on RSI perception, while uniform compensation without consideration of gaze information cannot control RSI effects at all.

- **Effect of superposition of spatially and temporally dependent compensation**: The effectiveness of compensation deteriorates for many subjects when spatially and temporally dependent compensation is combined. It is necessary to consider space × time interactions for the precise reduction of optical illusion, as spatial and temporal parameters interact with each other.

As described elsewhere, two main mechanisms of RSI related to eye movement have been proposed: refreshing the retinal image through large eye movements and updating the image with small eye movements. Our study dealt only with the former under gaze-free conditions, and the results confirmed that taking large eye movements into account would be effective for compensating for RSI perception. The results also indicated that it would be possible to compensate for RSI perception by physically rotating the image in the opposite direction of the perceptual motion, a conclusion that is consistent with previous studies [12, 14]. On the other hand, the results of Experiment 2 showed that the illusory effect was halved by 500 ms, while larger absolute PSE values were observed in the illusory condition than in the control setting,

even at longer compensation time. This suggests that not only to temporally dependent components derive from large eye movements but also that temporally independent components deriving from small eye movements likely exist, which is consistent with the results of Backus et al. [12] which reported a temporal dependence of the RSI perception in the eye-fixation condition.

In addition, we discuss the superposition effects observed in our experiment, namely, that the algorithm taking into account both spatial and temporal dependence (Algorithm C) had lower performance than algorithms considering either one or the other (Algorithms A and B). The reason for this may be that the tuning and selection of the compensation parameters were insufficient in both the calibration and the evaluation steps. Algorithm C compensated for this illusion by temporary rotation only to the peripheral vision alone. Thus, the subjects here were sometimes unable to adjust the parameters sufficiently to clearly distinguish the difference between two images. It was confirmed that Algorithms A, B, and C were more effective than Algorithm U, which suggests that taking gaze information into consideration is essential for RSI compensation, and the effects of the superposition of spatial and temporal dependence requires further investigation through the imposition of restrictions on the parameters and observation times.

Next, the possible application of the proposed system and algorithm is described. The fundamental points in the development of our system can be applied to other dynamic illusions that depend on gaze information. The system and algorithm can be applied to Fraser–Wilcox illusion groups with a similar pattern to that of RSI. These illusion groups cause perceived motion in still images, where the direction of motion is determined by the arrangement or brightness of the pattern. For example, the pattern used by Chi et al.[30], whose research developed a method to calculate the trajectory of a streamline using the Fraser–Wilcox illusion group, can be reduced by moving the given image against the perceived flow.

Further, other gaze-dependent illusions, including the Ouchi illusion, the Pinna illusion, the Hermann grid illusion, and the scintillation grid illusion, can be controlled using our system, although a compensation algorithm would need to be devised that takes into account the perceptual characteristics of each illusion. The Ouchi illusion exhibits a different direction of motion for images with check patterns, so image motion must be considered for its compensation. Furthermore, the Hermann grid illusion entails the perception of gray spots in a pattern consisting of black squares and white grids, compensation for which requires the consideration of the spatial dependence of the illusion.

As described above, the dynamic perceptive compensation proposed in this study can fundamentally be applied to other illusions, while separate algorithm development and system adjustment are required for each illusion.

## Limitations and challenges

This section describes the limitations and challenges encountered in this study.

First, the optimal parameters of the compensation system may change as the number of RSI disks and the brightness of the display change. This is because the algorithm parameter is dependent on specific pattern arrangement and brightness. Several parameters, such as the compensation angular velocity, would require the calibration phase according to the environment and displayed images.

In addition, it is difficult to apply dynamic visual compensation based on gaze information for some optical illusions. For example, the main effects of geometrical illusions such as the vertical-horizontal illusions or the Ebbinghaus illusions depend on cues other than gaze information, so different compensation methods are needed. These illusions can be compensated for by

enlarging or reducing the figures themselves [31]. Similarly, many illusions related to color and brightness, such as White's illusion [32], cannot be reduced with our proposed system. One solution for reducing these illusions has already been produced by adjusting the statistics using the gray world hypothesis or adjusting the white balance using machine learning [33, 34].

Therefore, it is essential to develop a more universal system. One possible solution would be to simulate the perception of dynamic illusions, including the RSI, on a computer. In mathematics and cognitive science, perceptual models that can explain illusion phenomena have been proposed. Not everything can be explained with a single model, of course, because optical illusions occur in complicated ways involving multiple levels of processing in the brain, while a model with a certain universality allows us to understand approximate human visual perception. The methods of constructing a perceptual model are broadly divided into a top-down approach that assesses visual information processing from a mathematical perspective [4, 5] and a bottom-up approach that generates human perception in a computer using machine learning [26, 35]. Both have succeeded in reproducing and generating illusions. In the context of our study, it may be possible to construct a model-based illusion compensation system by creating a reproduction model of illusions, including the RSI, for which some parameters can be experimentally determined.

Finally, we discuss how our proposed methodology, dynamic perceptive compensation synchronized with eye movements, can be used in various fields, such as VR and HCI. The fundamental operations of the measurement of eye movements, image manipulation, and compensation for illusory perceptions in this study can be utilized in relation to gaze-dependent perception. For example, image manipulation by suppressing optical flows that exist in peripheral vision will probably help maintain the sense of balance, which may lead to a reduction in motion sickness. In relation to compensation methodology, the results of our study suggest that it is necessary to account for both spatial and temporal dependence on eye movements. In other words, compensation should be based on the spatio-temporal characteristics of perception in relation to eye movements. Furthermore, our results suggest that it may be necessary to optimize the algorithm parameters using the superposition effect described above. Multi-dimensional optimization methods such as QUEST+ [36] will likely be an important part of effectively improving system optimization.

## Conclusion and future prospects

This study developed a dynamic perspective compensation system for the RSI synchronized with eye movements and evaluated its effectiveness in several experiments. The results obtained in this study are described below.

We developed a system to dynamically control optical illusions using an image presentation system that is synchronized with eye movements and an algorithm that took the characteristics of the illusion into consideration. In this study, the compensation system used took spatial dependence, temporal dependence, and individual differences into account. A dynamic system incorporating an eye tracking device enabled us to consider the spatial and temporal dependence of gaze information, which were difficult to achieve in the gaze constraints of previous studies. A preliminary experiment using paired comparison showed that the compensation with the proposed method reduced the effects of the RSI by setting appropriate parameters to our experimental conditions; the compensation angular velocity at 0.5 deg./s against the perceived motion reduced RSI effects, although the optimal parameter could vary in relation to the experimental condition. Moreover, the temporally dependent function of RSI perception was obtained in a further experiment, which suggested that short-term compensation of up to 500 ms was effective for RSI compensation.

Drawing on these considerations, the effects of spatially and temporally dependent gaze information on RSI compensation were evaluated in the paired comparison. The results for the algorithm with and without gaze information had a significant difference, with $p < 0.01$, calculated with the Bonferroni correction. The score values for some algorithms that took eye movements into account were higher than those for still images. These results indicated that compensation took gaze information into account was essential for reducing RSI effects. Both temporal and spatial dependences had an effect on RSI perception for the majority of subjects, while uniform compensation could not control RSI effects at all.

Consequently, we created a system that could control the perception of the RSI depending on gaze information, and this system was personalized. The results of our experiments indicated that the effects of the RSI could be diminished even at the speed of consumer devices, which suggests the possibility of widespread use of our technology at a low cost. However, a more precise compensation system could be realized through the improvement of real-time performance owing to a more precise detection of saccades. Studies on optical illusion presentation using head-mounted displays and projectors will also be important. Illusion perception can depend on presentation frequency and the characteristics of the video presentation device. The knowledge obtained in this study on dynamic perceptive compensation using eye tracking may prove to be applicable to the development of a video presentation system that eliminates perceptual discrepancies in augmented and virtual reality (AR / VR). Dynamic perceptive compensation focusing on optical illusions remains largely unexplored as a topic, and various studies are expected to provide insights into the mechanisms of the perception of illusions and guidance for engineering applications.

## Supporting information

**S1 Dataset. Dataset of eye tracking log data in the pilot study.** Data from a pilot study to investigate the sampling rate of eye tracking, provided in CSV format.
(CSV)

**S2 Dataset. Dataset of Experiments 1 to 3.** All data obtained in Experiments 1 through 3 are included. All data are available in the tabs in the Excel file.
(XLSX)

**S3 Dataset. Dataset of eye tracking log data in Experiments 1 to 3.** All data obtained in Experiments 1 through 3 are included. All data are available in tabs in the Excel file.
(XLSX)

**S1 Fig. Examples of gaze heat map obtained in our experiments without translucent stimulus image.** Heat maps for one subject (not the same subject across experiments) are shown for (a) Experiment 1, (b) Experiment 2, and (c) Experiment 3.
(EPS)

**S2 Fig. Examples of gaze heat map obtained in our experiments with translucent stimulus image.** Heat maps for one subject (not the same subject across experiments) with translucent stimulus image are shown for (a) Experiment 1, (b) Experiment 2, and (c) Experiment 3.
(EPS)

**S3 Fig. Gaze heat map of all subjects obtained in our experiments.** Heat maps for all subjects are shown for (a) Experiment 1, (b) Experiment 2, and (c) Experiment 3.
(EPS)

## Acknowledgments

The authors would like to thank the anonymous referees for their valuable comments and helpful suggestions. We also deeply appreciate the cooperation of the subjects who participated in the experiments.

## Author Contributions

**Conceptualization:** Yuki Kubota, Tomohiko Hayakawa, Masatoshi Ishikawa.

**Data curation:** Yuki Kubota.

**Formal analysis:** Yuki Kubota, Tomohiko Hayakawa.

**Funding acquisition:** Tomohiko Hayakawa, Masatoshi Ishikawa.

**Investigation:** Yuki Kubota, Tomohiko Hayakawa.

**Methodology:** Yuki Kubota, Tomohiko Hayakawa.

**Project administration:** Yuki Kubota, Tomohiko Hayakawa.

**Resources:** Yuki Kubota.

**Software:** Yuki Kubota.

**Supervision:** Tomohiko Hayakawa, Masatoshi Ishikawa.

**Validation:** Yuki Kubota.

**Visualization:** Yuki Kubota.

**Writing – original draft:** Yuki Kubota, Tomohiko Hayakawa.

**Writing – review & editing:** Yuki Kubota, Tomohiko Hayakawa, Masatoshi Ishikawa.

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
