## [Decision Letter · Decision Letter 0]

9 Dec 2020

PONE-D-20-21363

Dynamic Perceptive Compensation for the Rotating Snakes Illusion with Eye Tracking

PLOS ONE

Dear Dr. Kubota,

Thank you for submitting your manuscript to PLOS ONE. After careful consideration, we feel that it has merit but does not fully meet PLOS ONE’s publication criteria as it currently stands. Therefore, we invite you to submit a revised version of the manuscript that addresses the points raised during the review process.

We look forward to receiving your revised manuscript.

Kind regards,

Susana Martinez-Conde

Academic Editor

PLOS ONE

Journal Requirements:

Reviewers' comments:

Reviewer's Responses to Questions

**Comments to the Author**

1. Is the manuscript technically sound, and do the data support the conclusions?

Reviewer #1: Partly

Reviewer #2: Yes

2. Has the statistical analysis been performed appropriately and rigorously? 

Reviewer #1: I Don't Know

Reviewer #2: Yes

3. Have the authors made all data underlying the findings in their manuscript fully available?

Reviewer #1: No

Reviewer #2: No

4. Is the manuscript presented in an intelligible fashion and written in standard English?

Reviewer #1: No

Reviewer #2: Yes

5. Review Comments to the Author

Reviewer #1: In a series of three experiments, the authors test whether a gaze contingent display can be used to reduce the perceived motion in the rotating snakes illusion. This is expected to work, given previous research indicating that eye movements increase the strength of the illusion. The motivations for this work are a bit unclear in the manuscript. Two potential mechanisms are presented in the introduction, but what has been learned from this work in terms of the mechanism of perception of the illusions is not specified in the discussion. And I think that the guidance for engineering applications is unclear here, unless the authors simply mean that gaze contingent compensation should be considered a tool that can decrease gaze-related illusions.

The findings are generally sound, but the conclusions are over-stated. In several places, the authors state results as though the optimizations are general to all occurrences of RSI. For example, Experiment 1 is described as finding the “optimal” parameters. Lines 204-206 say that a ~.5 deg/s rotation compensation reduces RSI effects. But this and other values may be specific to the tested individuals and experimental setup (display brightness, specific pattern arrangement, lighting, etc.). These statements should be weakened. Further, as the authors point out, the effect is dependent on age and probably also a host of other individual factors. Thus, where the authors say the system “can be personalized”, it is probably more appropriate to say “must be personalized”. The statement that the established parameters “realizes to develop more universal compensation system for optical illusions” (lines 506-7) also seems to overstep the data: this system is specific to RSI.

I don’t understand how this study relates to visual perceptual discrepancies in VR and HCI (lines 41-46). If an observer perceives motion when presented with this kind of visual pattern, even though the image is still, this is natural vision. Illusions are perceived in normal vision. Under what VR or HCI situation would you want to augment vision by reducing visual illusions in the periphery—particularly if this process requires a great deal of calibration for specific people and contexts?

Line 133 says “Both systems were set to drive at 30 Hz” and it is unclear what two systems are referred to here. Were the display and eye tracker set to 30 Hz? Why are 90 Hz and 60 Hz values also reported in the same paragraph? The current text is unclear. A 90 Hz sampling rate is well below the benchmarks achieved by research-grade eyetrackers, and may negatively impact the saccade detection. However, a 30 Hz rate is definitely too low to be valid: previous work has shown that such low-resolution eye tracking fails to reproduce expected saccadic parameters.

Lines 228-231 suggest that the system compensated regardless of where participants looked on the display, but this was not specifically tested. True, participants were allowed to gaze freely across the screen. However, the authors assessed compensation as an average across trials, not on the basis of where participants were looking during a trial. The compensation could work better or worse in certain regions of the visual field.

Lines 305-307 say that selection bias “probably occurred”. The text says “half of the subjects looked at the non-illusory image after the illusory image” but was this tested? This could be assessed using the eye tracking data that was collected. If it was not tested, please specify that where participants looked—and in what sequence—is speculation.

The authors say that an “effect of superposition” may have prevented the combination of spatial and temporal compensation from being effective, but the authors do not describe this superposition effect (beyond describing the data). What mechanisms do you believe are in play here? Also, the text here says “However, the effect is still majorly higher than the still image even in that case” but algorithms O and C did not have a significant difference—and therefore C is not “majorly higher”. The combined compensation algorithm does not appear to be effective, and this should be clear in the discussion.

Minor:

While the manuscript is sufficiently detailed, the English language should be improved to ensure that an international audience can clearly understand the text. The current phrasing makes comprehension difficult throughout.

The Supporting Information contains subject-level averages, rather than raw data (trial-by-trial responses). As per PLOS data policy, the data points behind means, medians and variance measures should be available.

I am interpreting the reported “0.1%” significance levels as p=.001. If this is not correct, please clarify these percentages.

Reviewer #2: Review on Kubota et al. “Dynamic perceptive compensation for the rotating snakes illusion with eye tracking”

Overall evaluation

The authors present three experiments on perceptual compensation in the rotating snakes illusion, elaborating earlier work by different authors (e.g., Murakami et al. [14), Otero-Millan et al. [16]). Key innovation in the paper is a dynamic compensation system synchronized with eye movements. The overall writing is clear and the research rationale is convincing.

General comments

1. The abstract would probably benefit from reducing length and making it more succinct.

2. The eye-tracking device samples data at relatively low rate (90 Hz), which is comparably small when compared to state-of-the-art vision science labs. Could the authors please comment on the potential impact on the results?

3. I understand the purpose of the current work is to propose and analyze dynamic perceptive compensation procedures, however, from impression is that the new results are not adequately discussed in comparison to the earlier findings. Please comment.

Specific comments and typos

- l. 2: The first sentence of the main text is unclear. In what sense is it reproduced?

- l. 29: “depends”

EOF

6. PLOS authors have the option to publish the peer review history of their article (what does this mean?). If published, this will include your full peer review and any attached files.

Reviewer #1: No

Reviewer #2: No

---

## [Author Response · Author response to Decision Letter 0]

27 Jan 2021

Dear Reviewers

Thank you very much for your constructive comments and for the understanding of our research that you show. 

We have tried to prepare manuscript and "response to reviewers" files carefully, and we would appreciate it if you could check whether your reviewed comments are reflected appropriately.

We would like to take this opportunity to express our sincere thanks to the reviewers who identified areas of the manuscript that needed corrections or modification. 

Sincerely, 

Yuki KUBOTA, the University of Tokyo.

---

## [Decision Letter · Decision Letter 1]

17 Feb 2021

Dynamic perceptive compensation for the rotating snakes illusion with eye tracking

PONE-D-20-21363R1

Dear Dr. Kubota,

We’re pleased to inform you that your manuscript has been judged scientifically suitable for publication and will be formally accepted for publication once it meets all outstanding technical requirements.

Kind regards,

Susana Martinez-Conde

Academic Editor

PLOS ONE

Additional Editor Comments (optional):

Reviewers' comments:

Reviewer's Responses to Questions

**Comments to the Author**

1. If the authors have adequately addressed your comments raised in a previous round of review and you feel that this manuscript is now acceptable for publication, you may indicate that here to bypass the “Comments to the Author” section, enter your conflict of interest statement in the “Confidential to Editor” section, and submit your "Accept" recommendation.

Reviewer #2: All comments have been addressed

2. Is the manuscript technically sound, and do the data support the conclusions?

Reviewer #2: Yes

3. Has the statistical analysis been performed appropriately and rigorously? 

Reviewer #2: Yes

4. Have the authors made all data underlying the findings in their manuscript fully available?

Reviewer #2: Yes

5. Is the manuscript presented in an intelligible fashion and written in standard English?

Reviewer #2: Yes

6. Review Comments to the Author

Reviewer #2: Thank you for addressing my concerns. I am happy with the revised version of the manuscript.

7. PLOS authors have the option to publish the peer review history of their article (what does this mean?). If published, this will include your full peer review and any attached files.

Reviewer #2: No

---

## [Editor Report · Acceptance letter]

22 Feb 2021

PONE-D-20-21363R1 

Dynamic perceptive compensation for the rotating snakes illusion with eye tracking 

Dear Dr. Kubota:

I'm pleased to inform you that your manuscript has been deemed suitable for publication in PLOS ONE. Congratulations! Your manuscript is now with our production department. 

Kind regards, 

on behalf of

Prof. Susana Martinez-Conde 

Academic Editor

PLOS ONE